# Positive Organizational Psychology: A Bibliometric Review and Science Mapping Analysis

**DOI:** 10.3390/ijerph18105222

**Published:** 2021-05-14

**Authors:** Beatriz Martín-del-Río, Marie-Carmen Neipp, Adrián García-Selva, Angel Solanes-Puchol

**Affiliations:** 1Department of Behavioural Sciences and Health, University of Miguel Hernández, Elche, 03202 Alicante, Spain; bmartin@umh.es (B.M.-d.-R.); adrian.garcias@umh.es (A.G.-S.); 2Department of Health Psychology, University of Miguel Hernández, Elche, 03202 Alicante, Spain; neipp@umh.es

**Keywords:** positive organizational psychology, bibliometric analysis, science mapping

## Abstract

Positive organizational psychology (POP) is a research area that focuses on the positive aspects of optimal functioning at work. Although consolidated and with a large volume of publications, no bibliometric analysis has been performed that allows knowing its high-level structure, developments, and distribution of knowledge since its origins. The objective is to analyze the 7181 articles published in POP on the Web of Science Core Collection (WoSCC). A retrospective bibliometric analysis and science mapping were performed. The title, authors, institutions, countries, scientific categories, journals, keywords, year, and citations were extracted from WoSCC. Impact factor, quartile, and country were collected from Journal Citation Reports (JCR) 2019. Authors were classified according to the proposal of Crane, and Bradford’s law was calculated. The results show that it is an area with more than 100 years of experience, divided into three stages of different productivity and visibility, highlighting a decrease in its visibility in recent years. With a multidisciplinary and international interest, psychology and business and economics stand out, especially in countries such as the United States, the United Kingdom, and the Netherlands. Four popular study topics emerged: well-being at work, positive leadership, work engagement, and psychological capital.

## 1. Introduction

Martin Seligman, in his speech as president of the American Association of Psychology in 1998, stated that positive psychology (PP) is the scientific study of optimal human functioning [1]. Years later, Linley, Joseph, Harrington, and Wood [2], in their work of 2006, analyzed different definitions given by referents of PP, integrating them into the following: “The scientific study of the possibilities of optimal human development, at the meta-psychological level, aims to theoretically reorient and restructure the imbalance existing in psychological research and practices, granting greater importance to the study of the positive aspects of human life experiences, integrating them with those that cause suffering and pain; at a level of pragmatic analysis, it addresses the means, process, and mechanisms that make it possible to achieve a higher quality of life and personal realization” [2] (p. 8).

In short, PP goes from focusing on what is wrong with people to what is right, that is, it applies the scientific method to study the positive experiences, strengths, and psychological resources of people, groups, and institutions to achieve optimal development and functioning. As a result, the main research topics addressed by PP are oriented towards subjective well-being (happiness); positive moods and emotions; sensory, intellectual, and aesthetic pleasures; strengths and virtues; healthy practices such as optimism, gratitude, meditation, physical exercise, or artistic expression; interests, skills, and achievements; positive personal relationships; positive institutions (educational, labor, political); or transcendence and the meaning of life, among others.

This orientation spread very quickly to different fields of psychology and other educational disciplines and professions [3,4]. In 2007, the International Positive Psychology Association was created in the United States, and the first World Congress of Positive Psychology was held in Philadelphia in 2013. In Europe, the European Society of Positive Psychology was created, after the 6th European Congress of Positive Psychology held in Moscow in June 2012.

In this context emerges positive organizational psychology, which intends to focus on the positive aspects of the optimal functioning of people at work. Salanova and her team define it as the scientific study of the optimal functioning of individuals and groups in organizations, the effective management of psychosocial well-being, and the development of organizations to make them healthier [5]. Thus, taking into account this perspective, a significant number of researchers have gone from emphasizing more the negative aspects of organizations (burnout, stress, deficiencies, and problems of employers and employees, among others) to focusing on the positive aspects (eustress, management of change, and strengths and psychological capacities for the development of and increase in professional performance) [3,6,7]. However, there is still no consensus within the scientific community about the definition and focus of POP [8]. Therefore, labels such as positive psychology at work, positive workplace, or positive or healthy organizations have sometimes been used [9].

In recent decades, there has been a very significant increase in research on the constructs of PP applied to the context of organizations. In the literature can be found different reviews that have been carried out from the broader areas of PP [10], well-being at work [11], and work and organizational psychology [12], or from more specific ones such as ethical behavior in leadership [13], happiness management [14], happy–productive workers [15], or the relationship between job crafting and healthy organizations [16], or even related to intensive-care nurses’ well-being [17]. 

Although the aforementioned review studies report important data for understanding how aspects of PP are being studied in POP, they do not provide a global overview of the development of this scientific domain. Moreover, they do not indicate which are the most important topics in this area, nor do they provide an overview of the general structure in terms of the impact of journals, institutions, authors, and its historic development and trends. Therefore, this paper aims to fill this gap, to allow researchers to understand the developments and knowledge distribution of this scientific research domain. For this purpose, a bibliometric analysis and mapping review are appropriate. These methodologies explore the nature and structure of scientific activities [18]. We think it would be important for experts and scholars who are interested in the POP to understand its current research status to be able to propose future research.

In this study, a bibliometric analysis was carried out to analyze the development of POP, including the temporal and geographical distribution of publications and citations and the distribution of scientific categories, and the most relevant journals in this domain and the most cited papers in the last 15 years are also reported. On the other hand, VOSviewer was applied to assess and visualize collaborative co-authors, research institution networks, and keyword co-occurrences.

The remainder of this paper is organized as follows: Section 2 covers the design of the study and the data source and introduces the methods used in the bibliometric analysis and the software used for the visual analytics. Section 3 describes the results, including temporal distribution of published articles and cited papers; scientific categories; geographical distribution of countries, institutions, and cooperation networks; collaborative co-author network analysis; most relevant journals in POP; analysis of most cited papers in the last 15 years; and keyword co-occurrence analysis. Section 4 presents the discussion, and conclusions and limitations are included in Section 5.

## 2. Materials and Methods

### 2.1. Design of the Study

An observational, descriptive retrospective study was conducted using bibliometric methodology and science mapping. As it does not include human beings, it did not need the approval of any institutional review board. 

### 2.2. Data Collection

On 7 February 2021, a thorough search was conducted in the WoSCC of the ISI Web of Science (Thomson Reuters, Philadelphia, PA, USA). The following terms and retrieval strategy were used: (positiv* AND organization*) *OR* (positive* AND organization* AND psychol*) *OR* (positive* AND organization* AND scholar*) *OR* (positive* AND psychol* AND work*) *OR* (positive* AND workplace) *OR* (positive* AND organization* AND behave*) *OR* (positive* AND human* AND resour*) *OR* (positive* AND leader*) *OR* (psychol* AND capital*) *OR* (organizat* AND virtuous*) *OR* (“organizat* resil*”) *OR* (“resil* organizat*”) *OR* (happy* AND work*) *OR* (happi* AND work*) *OR* (well-bein* AND work*) *OR* (wellbeing* AND work*) *OR* (“flow* at work*”) *OR* (cope* AND work* AND stress*) *OR* (coping* AND work* AND stress*) *OR* (work* AND engage*) *OR* (organizat* AND ethic*). The searches were conducted in the title field. These terms were chosen from among those obtained as a result of the review of Donaldson and Ko [19] on the status of the POP area between 2001 and 2009. 

A total of 10403 works were obtained (Figure 1). We used the following inclusion and exclusion criteria: (1) there were no restrictions on language or availability; (2) works published in 2021 were not included; (3) according to the document type, only the article, article early-access, review, and review early-access were selected. Finally, 7181 articles were included in the bibliometric analysis.

### 2.3. Data Analysis

WoSCC was used to obtain data on the years of publication of the works, the citations received, the scientific categories in which the journal that published the work is classified, the authors and their affiliation (institution or organization and country), the journals, and the keywords provided by the authors and the KeyWords Plus automatically generated from the titles of the articles. From the Journal Citation Report (JCR) edition 2019, the data related to the journals were obtained, such as the impact factor, the thematic category of each journal, the quartile of each journal in its category, and the country of the publisher. We followed the proposal of Crane [20] for the classification of authors according to their volume of publications, in aspirants (1–4 papers published in POP), moderate producers (4–10 papers), and high producers (more than 10 papers). Moreover, the law of Bradford [21,22] was calculated to determine the objective weight of each journal, as it distributes the journals into three productivity zones with a similar number of articles and an increasing number of journals.

Microsoft Excel 2018 was used to conduct the descriptive study of the years of publication, citations received, scientific categories, authors and signatures, affiliations, countries, and journals. This software also allowed calculating the growth prediction of the publications until 2024, as well as creating the bubble plot of the temporal distribution of the works and citations received according to the scientific category. The tool MapInSeconds (mapinseconds.com) (accessed on 3 March 2021) of Darkhorse Analytics Inc. was used for the representation of the global geographical distribution of the articles.

VOSviewer 1.6.15 [23] was used to extract information, obtain the clusters, create network maps on cooperation by country in the publication of articles, establish the collaborative network between authors, and perform the keyword co-occurrence analysis. Prior to the completion of each network, the data were reviewed and standardized manually so that each country, author, or keyword had a single written form. In the network interpretation, each node represents a country, author, or keyword. The size of each node indicates its frequency, so that the larger it is, the higher the occurrence. The line that joins the nodes represents the existence of a connection between countries, authors, or keywords. The thicker these lines are, the greater the relationship between the nodes. Finally, the colors of the nodes represent the clusters of countries, authors, or keywords, such that those of the same color belong to the same cluster. 

## 3. Results

### 3.1. Temporal Distribution of Published Articles and Cited Papers

The 7181 articles obtained in the search for POP were published over 116 years and have been cited 174,158 times. Of these, 41 can be considered citation classics or highly cited papers, as they received 400 or more citations, and most of them (80.50%) were published over a period of 10 years (2001–2011).

Three time periods are observed (Figure 2). The first one, the incubation period (1904–1994), begins in 1904 with the work of Betty Eicke and Anne S. Bussell “The shortcomings of the teaching methods of the present training-schools for nurses from the standpoint of the graduate nurse engaged in private work” [24]. During these 90 years, 369 works (*M =* 4.10 papers/year) were published irregularly with 18925 citations (*M* = 51.29 citations/work). The most cited article of William A. Khan, “Psychological conditions of personal engagement and disengagement at work” [25] (2800 citations), belongs to this period.

The second period, the initiation (1995–2007), reveals the first increase in the number of works (*n* = 932) published on a regular and incremental basis (*M =* 71.69 works/year), which received a total of 66635 citations (*M =* 71.50 citations/work). As of 2008, a third time period of exponential growth (2008–2020) of 13 years was also initiated, with 5880 articles (*M* = 452.31 works/year). When focusing on the prediction for the next four years, the exponential increase observed in the last stage is confirmed, revealing its maximum real value in 2020 with 1136 works. However, although 88,598 citations were received in the latter period, the highest value of the three, the average number of citations per work was 15.07, much lower than the previous ones.

### 3.2. Scientific Categories

The classification of each article into a scientific category was based on the classification carried out by the WoSCC. If a work belonged to multiple categories, it was multi-assigned, so the final count would be higher than that of the articles in the initial search.

POP articles were classified into 132 different thematic categories. Figure 3 shows the temporal evolution of the works and citations in categories with 50 or more works. “Business and Economics” is the area where the largest number of works were published (*n* = 2179), with a broad time-span since 1962 and with sustained growth. It is also the area that received the highest number of citations (*n* = 82,145; 37.70 citations/work), showing a marked decline since 2006. Secondly, the “Psychology” setting presents 1995 articles and a development similar to that of the previous area. It has a broader time-span (since 1959), a steady increase in the number of works and citations (*n* = 80,175 citations; 40.19 citations/work), and a considerable reduction in its volume of citations as of 2006. 62% of the citation classics or highly cited papers belong to these two categories.

The results show that the POP area is very interdisciplinary and with variations over time. During the incubation period, the works were published in 70 scientific categories, highlighting “Psychology” (94), “Business and Economics” (86), “Public, Environmental, and Occupational Health” (39), “Social Sciences—Other Topics” (38), and “General and Internal Medicine” (34). In the initiation stage, there were 86, also with “Psychology” (239), “Business and Economics” (236), “Social Sciences—Other Topics” (135), and “Public, Environmental and Occupational Health” (72) at the top. In the exponential growth stage, the number of categories increases to 122, again with the highest volume of publications in “Business and Economics” (1857), “Psychology” (1662), “Public, Environmental, and Occupational Health” (700), and “Social Sciences—Other Topics” (637).

### 3.3. Geographical Distribution of Countries and Institutions and Cooperation Network

The works on POP involved 123 countries on the five continents (Figure 4) although only 60 countries contributed more than 10 works. The 10 countries that made the highest number of contributions were the United States (1895) and Canada (374) in North America; China (569) in Asia; the Netherlands (462), Spain (323), the United Kingdom (639), Germany (299), and Italy (207) in Europe; South Africa (236) in Africa; and Australia (513) in Oceania. Regarding the impact of these works on the scientific community, the countries that received the highest number of total citations were the United States (63,697), the Netherlands (27,622), the United Kingdom (13,722), Canada (11,230), and Australia (10,351). However, the three countries that received the highest number of work citations were the Netherlands (59.79), Finland (44.47), and the United States (33.61).

The analysis of international participation over time (Table 1) shows that in the first 90 years (incubation period), the works were signed by authors from 16 countries, with the United States being the most productive and most cited. In the initiation period, the number of countries increased to 46, and the United States also led the productivity and number of citations. Finally, during the period of exponential growth, up to 122 countries were included. The United States also contributed the most articles and citations. The top five of international participation included the United Kingdom in all three time periods, Canada in the initial two, and the Netherlands in the last two.

With regard to international cooperation (Figure 5), there are 15 collaborative groups. The United States shows the largest collaborative network, with 1895 articles published by 78 different countries, including China (77), Canada (71), the United Kingdom (55), Australia (47), and South Korea (45). It is followed by the United Kingdom, with 639 works with 67 countries, among which the United States (55), Australia (49), the Netherlands (31), Germany (31), and China (26) stand out. Third, China participates in 569 works together with 50 countries, most frequently collaborating with the United States (77), Australia (47), Taiwan (28), the United Kingdom (26), and the Netherlands (17).

A total of 1000 institutions signed these works. Fifty-four percent did so with five or fewer participations. The 10 most productive were Utrecht University in the Netherlands (127), Erasmus University in the Netherlands (122), North-West University in South Africa (76), Katholieke University of Leuven of Belgium (63), Pennsylvania State University in the United States (59), the University of Michigan in the United States (58), the University of Jyväskylän in Finland (56), Monash University in Australia (53), the University of Valencia in Spain (52), and the University of North Carolina in the United States (51).

### 3.4. Collaborative Co-Author Network Analysis

These publications involved 17,831 different authors, an average of 2.48 authors per article. Following the proposal of Crane [20], 17,438 authors are aspirants because they have participated in less than four works, 393 are moderate producers, as they have signed between 4 and 10 papers, and 38 are high producers, with more than 10 works.

When analyzing the moderate and high producers (Figure 6), 70 clusters or collaboration groups were obtained. The one that groups the largest number of documents and collaborations is also the largest, with 28 authors collaborating with 185 authors, signing 308 documents that received 33,440 citations. Arnold B. Bakker (93), Evangelia Demerouti (35), Sabine Sonnentag (21), Toon W. Taris (19), and Despoina Xanthopoulou (13) in the Netherlands and Greece stand out in this group as high producers.

The second group involves 15 authors who sign with 61 authors in 154 documents that have been cited 23,246 times. Fred Luthans (35), Marisa Salanova (31), James B. Avey (16), and Susana Llorens (11), in the United States and Spain, are the high producers. The third group is made up of seven authors who participate in 114 works with 53 authors, receiving 13811 citations. Wilmar Schaufeli (82) from the Netherlands leads this group. The fourth group is made up of 12 authors who signed with 107 authors and received 9356 citations. Ulla Kinnunen (30), Jari J. Hakanen (18), Taru Feldt (16), Saija Mauno (14), Jessica de Bloom (13), and Asko Tolvanen (11) are at the forefront of the productivity of this Finnish group. 

### 3.5. Most Relevant Journals in POP

The works were published in 2320 different journals. When applying the law of Barford (1934), a distribution in three areas was obtained. The first or core area includes 64 journals that published between 236 and 16 papers, the second area includes 490 journals that published between 15 and 3 articles, and the third area includes 766 journals that published 1 or 2 papers.

The analysis of the 64 journals of the core area (Table 2 presents the data from the 20 most productive journals in the core area) shows that 29 were published in the United Kingdom; 20 in the United States; 4 in the Netherlands; 2 in South Africa and Germany; and 1 in Japan, Romania, and Spain. Most are related to the areas of “Psychology,” “Business and Economics,” and “Public, Environmental and Occupational Health” (65.52%). The impact factor ranges from 0.453 to 6.941 (mean 2.34 ± 1.29). Most (68.42%) are in the first and second quartiles of their category. 

### 3.6. Most Cited Paper Analysis in the Last 15 Years

Table 3 shows the articles that have received the highest number of citations in the years corresponding to the initiation and exponential growth periods (1995–2020). Together they received 8.56% of the total citations. They are published in 20 different journals, mostly (90%) located in quartiles 1 and 2 of their category, belonging to the categories of “Psychology” (44.74%) and “Business and Economics” (42.11%). Their authors are mostly from the United States (38.99%), the Netherlands (16.67%), and Spain (8.33%). Concerning their study design, most are cross-sectional, which use a multivariate methodology for verifying the fit of causal models (15). The reviews, especially theoretical reviews and discussion of topics (7), are also noteworthy. To a lesser extent, three works of instrument creation or validation and one longitudinal work are included.

### 3.7. Keyword Co-Occurrence Analysis

An analysis of the 16,555 author keywords and KeyWords Plus included in the 7181 articles was carried out. Of them, 81.90% only appear once or twice. When performing a co-word cluster analysis with terms that have a frequency of 40 or higher, four clusters were obtained that are represented in Figure 7 from left to right. The keywords were reviewed and modified so that each meaning was represented by a single written form (for example, well-being instead of wellbeing).

Cluster 1 includes 81 keywords with 10,654 occurrences. It is related to well-being at work, and its main terms are stress (756), health (754), well-being (629), mental health (390), social support (302), depression (280), personality (273), validation (273), nurses (256), happiness (235), gender (225), support (200), predictors (197), intervention (191), life satisfaction (189), life (187), occupational stress (171), care (169), experience (168), and conflict (154). Cluster 2 consists of 75 terms with 11844 occurrences; it is related to positive leadership and is represented by keywords with a high occurrence such as performance (1286), job-satisfaction (915), model (736), behavior (580), antecedents (446), work (389), management (365), commitment (355), mediating role (334), leadership (312), attitudes (295), workplace (279), perceptions (251), transformational leadership (220), ethics (207), organizational commitment (188), turnover intention (175), organization (163), moderating role (159), and self (150).

Cluster 3 is about work engagement; it includes 35 keywords with 8192 occurrences. Those that have the highest occurrence are work engagement (1188), burnout (1168), resources (759), satisfaction (668), job demands (512), engagement (395), employee engagement (311), demands (310), employees (302), motivation (256), job (177), job resources (173), questionnaire (172), conservation (161), demands-resources model (152), psychological conditions (136), fit indexes (129), self-determination theory (115), mediation (109), and positive emotions (97). Finally, Cluster 4 includes 19 terms with an occurrence of 3311, and relates to psychological capital, with terms such as impact (770), psychological capital (428), self-efficacy (349), meta-analysis (270), job-performance (206), positive psychology (192), resilience (189), optimism (115), emotions (108), hope (95), psychology (81), positive organizational-behavior (74), employee performance (70), efficacy (67), emotional intelligence (65), organizational behavior (62), individual-differences (58), construct (57), and core self-evaluations (55).

## 4. Discussion

The objective of this work was to carry out a comprehensive analysis of the evolution of research publications in the field of POP. A bibliometric methodology and science mapping were used in order to analyze publication and citation trends, the scientific categories, the top players in terms of journals and institutions, the collaborative networks between authors and countries, and the central research topics. Moreover, in this section, we discuss research gaps and some promising scientific directions for future research in the POP field, which can be shown as follows:The field of POPs has evolved in three main stages over time and can be considered interdisciplinary, mainly around two predominant areas of research. The 7181 articles analyzed were published over a broad 116-year period, in which three temporal periods can be distinguished: one more extensive period, between 1904 and 1994, with a very irregular frequency of publication; a second period of initiation between 1995 and 2007, with a progressive increase; and a third period of exponential growth, from 2008 to the present moment. The same trend has been observed in the study by Sott et al. [12] in their review of 100 years of scientific evolution in the field of work and organizational psychology. As of this third stage, interest in a proactive and positive approach to the study of organizations is consolidated, which, as indicated by Luthans [7], focuses more on strengths, that is, what is right in organizations, teams, leaders, and employees rather than on what is wrong. This interest, which arises from the current PP that began in the 2000s, quickly spread to professionals and academics in the field of management and organization, as well as in industrial and organizational psychology.

This fact is confirmed by the results of this study on the analysis of scientific areas and journals. Although it is an area of multidisciplinary study with variations throughout its century of existence, the two areas that are present since its inception and that have published together 43.54% of the works on POP are those of “Psychology” and “Business and Economics.” This is also confirmed in the results of the journals because, according to Bradford [21], 13 of the 20 that make up the core zone belong to these two areas. This is consistent with the definition of POP [6], as it covers both organizational and work aspects and psychological aspects.

Scholars and experts can explore new categories from other perspectives, such as mathematics or computer sciences, to acquire more meaningful conclusions. Although this category does not appear in Figure 3, mathematics and computer sciences may be a potential research opportunity in the future. For example, the topic of people analytics is future-oriented. In recent years, there has been talk of the future of work, with stories of all jobs being replaced by robots and automated processes. Automation is here and it will impact the job landscape, but the key point is that with automation comes data, and so for us to ignore this new source of data is foolhardy. Digital technologies can provide new sources of people insight and new ways of collaborating across an organization and help with resolving challenges and issues that had not been considered before [51]. The emergence of new technologies is important for researchers and scholars to improve the precision of prediction of organizational behavior.

The field of POP has grown and continues to grow exponentially, but for a few years, this growth has not been accompanied by the number of citations these papers have received. Concerning the analysis of citations carried out, it should be noted that their amount has been increasing in the first two periods of publication. This is an objective measure of performance, value, recognition, influence, and impact of an investigation and its researchers [52]. However, the exponential increase in the third period was not matched by an increase in the number of citations received, but rather the contrary. Between 2012 and 2020, this number has been declining dramatically, going from 34.26 citations per work in 2012 to 1.19 in 2020.

As indicated by Garfield in 1977 [53], the reasons for citing a publication are varied. These would include providing a background reading, identifying a methodology, or identifying an original publication describing concepts or discussing related ideas, among others. In any case, we consider it necessary to deepen the study of the evolution of the aforementioned works in the POP area as of 2012 to be able to explain why a decrease in the number of citations and an increase in the number of works are produced. 

Firstly, there is evidence for and against the greater likelihood of the oldest works being cited [54]. In addition, it also appears that those published in the last 15 years have less chance of becoming citation classics or highly cited papers in their area [55]. In the case of POP, these two aspects are confirmed. Most of the citation classics, those that have generated high interest in the scientific community, were published between 2001 and 2011. This fact is confirmed because, to this date, the most cited articles each year (except for four of them) received more than 400 citations, highlighting “The measurement of work engagement with a short questionnaire—A cross-national study” [36] with 2386 citations and the work of Luthans and collaborators “Positive psychological capital: Measurement and relationship with performance and satisfaction” [37] with 1220 citations. Therefore, we could conjecture that this last stage could be the frontier of knowledge in POP. As Cole indicated [56,57], in the scientific progress of an area, a distinction can be made between core and frontier knowledge. The core includes the fundamental theories of a field, whereas the more ephemeral frontier knowledge is produced in recent years, including descriptive analyses with little temporal transcendence, and it does not function as a basis for further knowledge development. It is important that academics and experts who research in the field of POP make their work visible to the scientific community.

Concerning international representation, although very extensive, with the participation of 1000 institutions from 123 countries on five continents, the analysis of the data shows that in each of the three time periods analyzed, the United States, the United Kingdom, and the Netherlands have contributed the most publications and received the most citations. They are also the ones with the greatest international collaboration, along with Australia, China, and Spain.

The United States is also the country that shows a greater collaborative network with countries such as China, Canada, the United Kingdom, Australia, and South Korea. Its high economic capacity and its institutions with a high level of research cause many researchers from other countries to be interested in establishing research links [58]. It has shown clear leadership in POP since its inception, with Fred Luthans from the University of Nebraska and James B. Avey from Central Washington University standing out among all the authors.

The Netherlands also stands out, with the teams led by Arnold B. Bakker from Erasmus University Rotterdam and Wilmar Schaufeli from Utrecht University. Although the United Kingdom is noteworthy for its productivity, it does not present any referent team in this field. Finally, although China has recently entered the scene with high scientific productivity and a wide collaborative network, it has not been accompanied by high visibility, with very low levels of citation. Therefore, as Zhu and Liu [59] indicate, China may need a change in its research evaluation standards and should focus on improving the quality of research. It would be interesting to the scientific community to increase international collaboration between countries and research groups to consolidate POP.

The results of the keywords co-occurrence analysis showed a distribution in four thematic areas, in addition to highlighting the most popular topics in POP. This is in line with the findings of Soot et al. [12] in defining them as new themes unrelated to earlier periods in the development of work and organizational psychology. The terms of the first cluster refer to the research topic of well-being at work, which can be understood as the quality of life at work, which is considered the greatest determinant of productivity at the individual, organizational, and social level [60]. The importance of well-being in the workplace can be seen from the positive outcomes (e.g., health, mental health, social support, happiness, life satisfaction, coping) of well-being and the negative impact (e.g., stress, depression, occupational stress, conflict, family conflict, negative affectivity) when it is not fulfilled [61]. Studies on instruments for evaluation and intervention in this area are also noteworthy.

The second cluster, related to positive leadership, includes terms such as transformational leadership, ethical leadership, and authentic leadership. These studies show the characteristics of positive leadership and its potential benefits for the leader, the employees, and the organization [19], such as positive organizational behaviors, management, improved performance, job satisfaction, or organizational commitment, among others.

Thirdly, the study of work engagement is mainly related to the job demands–resources model [62] and the concept of burnout [63], with terms such as burnout, resources, job resources, demands, job demands, and demands–resources model. Lastly is the research on psychological capital. Following the definition of Luthans, Avey, Avolio, Norman, and Combs [64], the keywords grouped in this cluster refer to the positive psychological state related to self-efficacy when assuming and making the necessary efforts to succeed in tasks of some difficulty, optimism about current and future success, hope of success and perseverance in the objectives, and resilience to achieve success in the face of problems and adversity. The emergence of new research topics, the combination of different approaches, and the determination of how to use different methodologies to discover more meaningful results in the prediction of POP can be viewed as an interesting direction for the future.

## 5. Conclusions

During the past few years, there have been no reviews about POP and no in-depth scientometrics analysis. This work analyzes the evolution of research in POP as of its origins, using a bibliometric methodology and science mapping. 

According to the above bibliometrics analysis, the main conclusions of this study can be summarized as follows: (1) this area of research began in 1904 and developed around three time periods, consolidating this positive approach in the study of organizations as of 2008; (2) its volume of publications, as well as its visibility, was increasing until 2011, at which point a significant decrease in the number of citations received began; (3) although it has had a multidisciplinary approach since its inception, it has been of particular interest for Psychology and Business and Economics; (4) it has generated research interest in 123 countries on five continents, with the United States, the United Kingdom, and the Netherlands standing out particularly, both for their productivity and for their international collaborative networks; (5) over the years, four research topics have been highlighted as the most popular: well-being at work, positive leadership, work engagement, and psychological capital.

This study presents some limitations. Only the WoSCC was used for the search, so the number and type of works, as well as the number of citations received, depend on it and would have been different if the search had been completed with searches in other databases such as Scopus or Google Scholar. Although the WoSCC does not include the entirety of peer-reviewed publications, it does allow access to extensive and multidisciplinary information from more than 250 sciences, with more than 21,100 peer-reviewed, high-quality scholarly journals, books, and conference proceedings published worldwide since 1800. It is also possible that some important articles with great influence in POP are not included in this work due to the search profile used, which, despite being very extensive, may have left out terms of great relevance to this area of knowledge. We also recognize the limitation of not taking into account the self-citations, or whether the citations received by the works are positive or negative, so the level of agreement or criticism of the scientific community towards the cited articles cannot be determined [65].

## Figures and Tables

**Figure 1 ijerph-18-05222-f001:**
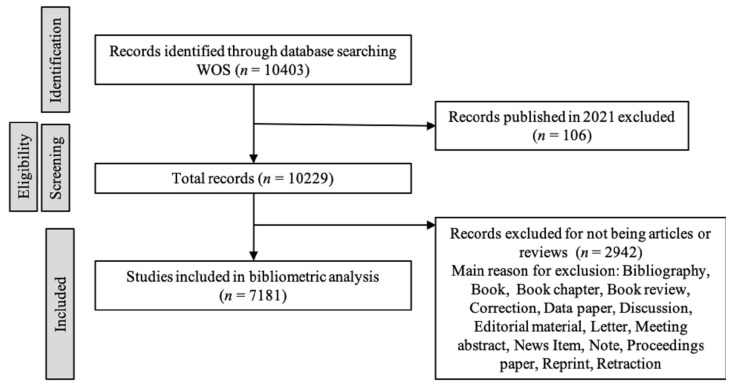
Flow chart of information of the retrieval strategy of articles on POP.

**Figure 2 ijerph-18-05222-f002:**
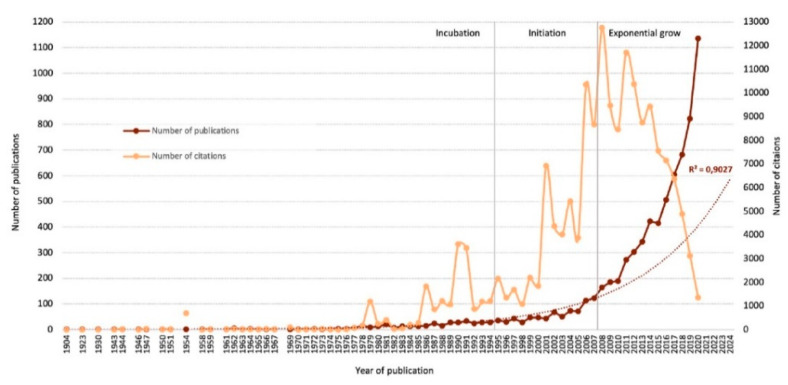
Temporal distribution of the published articles and citations received in POP and growth prediction for the next four years.

**Figure 3 ijerph-18-05222-f003:**
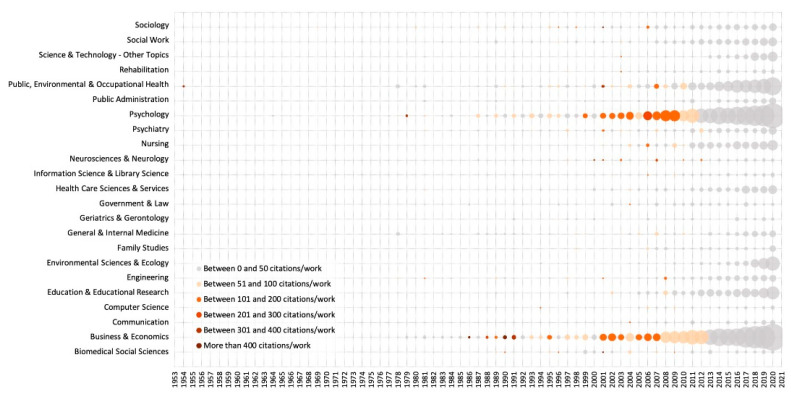
WoSCC scientific categories in POP research. Note: X-Axis: Publication year, Y-Axis: Scientific category, Bubble Size: number of works, Colors: citations/work.

**Figure 4 ijerph-18-05222-f004:**
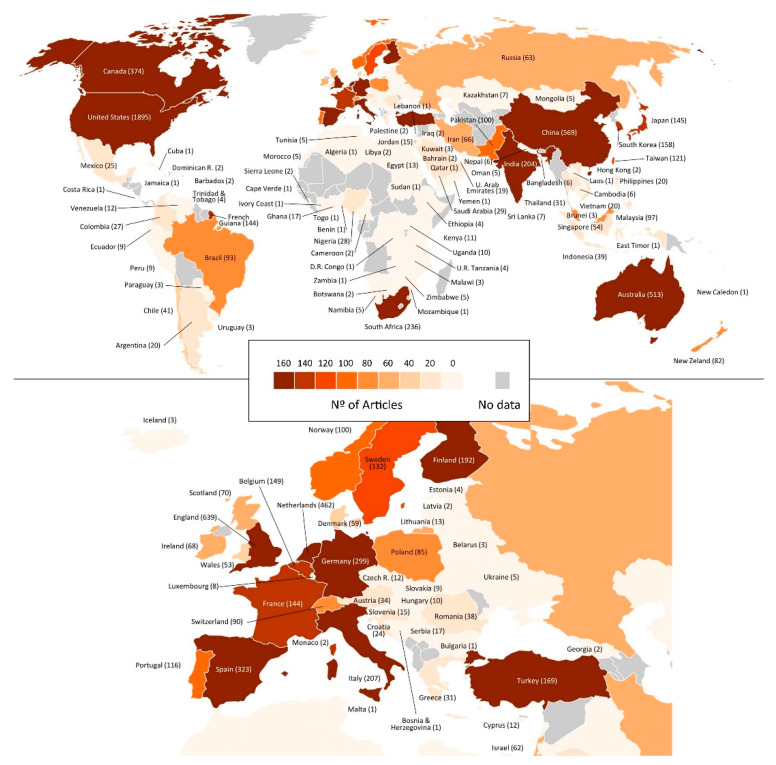
Geographical distribution of articles published in POP.

**Figure 5 ijerph-18-05222-f005:**
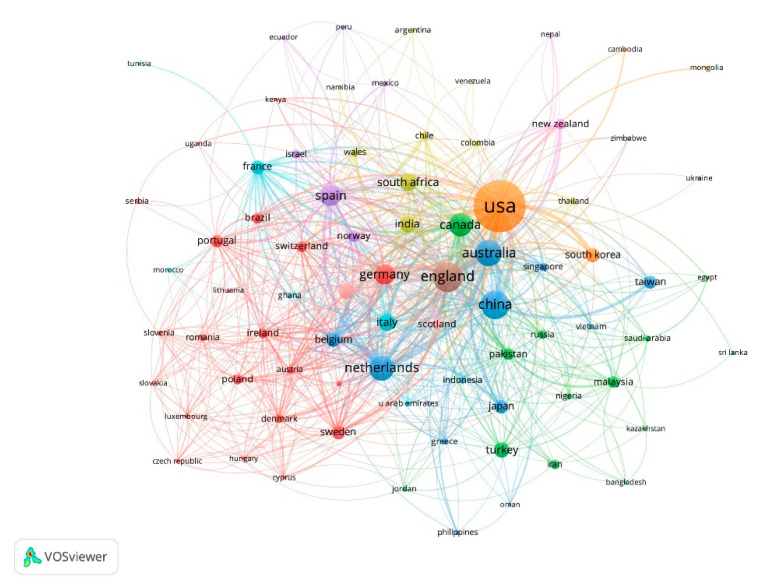
Network country cooperation based on co-authorship.

**Figure 6 ijerph-18-05222-f006:**
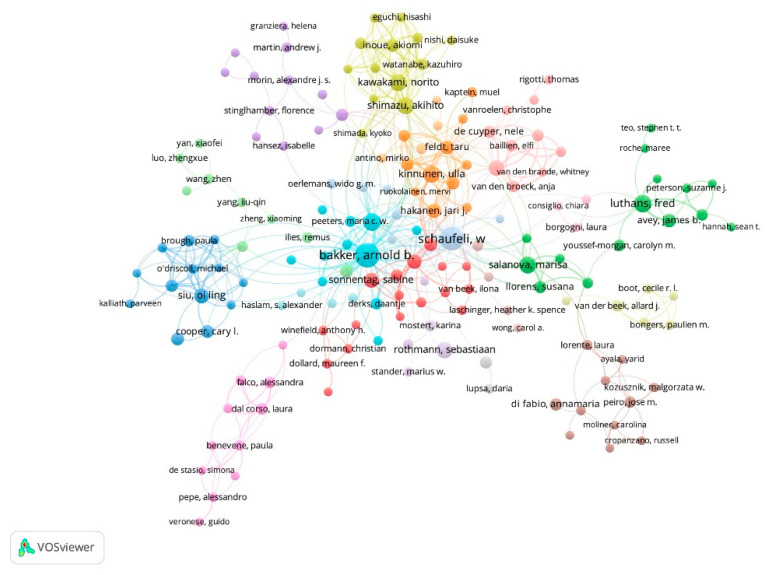
Clusters and co-authorship network in POP for authors who have published four or more papers.

**Figure 7 ijerph-18-05222-f007:**
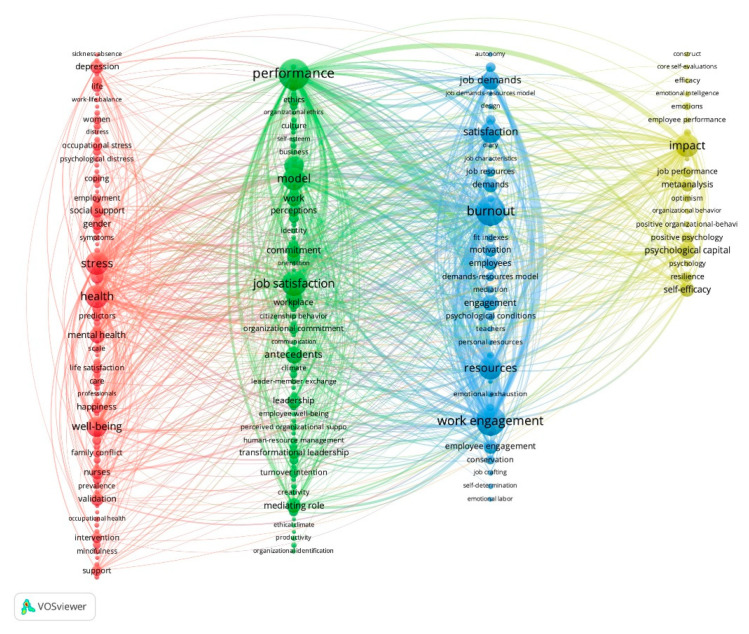
Diagram of keyword clusters.

**Table 1 ijerph-18-05222-t001:** Top 5 countries, papers, and citations in the three temporal periods.

	**Incubation**			**Initiation**			**Exponential Growth**		
Countries	16			46			122		
Articles	137			788			8044		
Citations	6357			60,325			137,942		
Citations/work	46.40			76.55			17.15		
**Top 5**	**Country**	**D ^1^**	**Cites**	**Country**	**D ^1^**	**Cites**	**Country**	**D ^1^**	**Cites**
	United States	78	5211	United States	341	26,464	United States	1476	32,022
	Russia	20	10	United Kingdom	69	3819	China	562	7940
	United Kingdom	8	249	Canada	58	3952	United Kingdom	562	9654
	Canada	7	173	The Netherlands	45	8516	Australia	490	9312
	Japan	7	39	Germany	31	2439	The Netherlands	416	19,095

^1^ Documents published.

**Table 2 ijerph-18-05222-t002:** Journals that have published the most works in POP.

Journal	D ^1^	Cites	C/W ^2^	IF ^3^	Q ^4^	WoSCC Category ^5^
*Journal of Business Ethics*	263	10,950	41.63	4.141	Q1	BUSS, ETH
*Frontiers in Psychology*	146	986	6.75	1.784	Q2	PSYC
*Inter. J. of Environ. Research and Public Health*	108	509	4.71	2.341	Q2	ESandE, PEOH
*J. of Vocational Behavior*	69	5355	77.61	3.396	Q1	PSYC
*Work and Stress*	67	5055	75.45	3.390	Q1	PSYC
*European J. of Work and Organizat. Psychol.*	64	2184	34.13	2.504	Q1	PSYC, BandE
*Inter. J. of Human Resource Management*	61	2002	32.82	2.908	Q2	BandE
*J. of Occupational Health Psychology*	59	3620	61.36	6.941	Q1	PSYC, PEOH
*Sustainability*	58	223	3.84	1.711	Q2	SandTOT, ESandE
*SA J. of Industrial Psychology ^6^*	51	956	18.75	-	-	PSYC
*J. of Happiness Studies*	49	1036	21.14	2.179	Q1	PSYC, SCOT
*J. of Occupational and Environmental Medicine*	46	659	14.33	1.436	Q3	PEOH
*J. of Occupational and Organizational Psychol.*	46	3563	77.46	2.545	Q2	PSYC, BandE
*Plos One*	45	661	14.69	2.650	Q2	SandTOT
*Human Relations*	45	2381	52.91	3.320	Q1	BandE, SCOT
*Social Indicators Research*	43	647	15.05	1.874	Q2	SCIN, SOC
*J. of Organizational Behavior*	44	5002	113.68	4.832	Q1	BandE, PSYC
*Personnel Review*	42	696	16.57	1.731	Q2	BandE, PSYC
*Social Behavior and Personality*	39	326	8.36	0.575	Q4	PSYC
*J. of Nursing Management*	37	817	22.08	1.773	Q1	BandE, NUR
First area of productivity	2394	71,367	29.81			
Second area of productivity	2611	26,053	9.98			
Third area of productivity	782	11,962	5.50			

^1^ Documents published. ^2^ Citations per work. ^3^ Impact Factor. ^4^ Quartile in its category. If a journal belongs to multiple categories, the one with the highest value is included. ^5^ BUSS: Business; BandE: Business and Economics; ESandE: Environmental Sciences and Ecology; ETH: Ethics; NUR: Nursing; PSYC: Psychology; PEOH: Public, Environmental, and Occupational Health; SandTOT: Science and Technology—Other Topics; SCIN: Social Sciences—Interdisciplinary; SCOT: Social Sciences—Other Topics; SOC: Sociology. ^6^ The *SA Journal of Industrial Psychology* presents no data in the JCR 2019 as it is not indexed on it.

**Table 3 ijerph-18-05222-t003:** Top cited papers between 1995 and 2020.

Year	Authors	Title	Journal	Citations
1995	Hosmer, L. T.	Trust—The connecting link between organizational theory and philosophical ethics [26]	*Academy of Management Review*	914
1996	Parasuraman, S., Purohit, Y. S., Godshalk, V. M., and Beutell, N. J.	Work and family variables, entrepreneurial career success, and psychological well-being [27]	*Journal of Vocational Behavior*	385
1997	Schaubroeck, J., and Merritt, D. E.	Divergent effects of job control on coping with work stressors: The key role of self-efficacy [28]	*Academy of Management Journal*	218
1998	De Jonge, J., and Schaufeli, W. B.	Job characteristics and employee well-being: a test of Warr’s Vitamin Model in health care workers using structural equation modelling [29]	*Journal of Organizational Behavior*	154
1999	Danna, K., and Griffin, R. W.	Health and well-being in the workplace: A review and synthesis of the literature [30]	*Journal of Management*	580
2000	Anshel, M. H.	A conceptual model and implications for coping with stressful events in police work [31]	*Criminal Justice and Behavior*	209
2001	Deci, E. L., Ryan, R. M., Gagne, M., Leone, D. R., Usunov, J., and Kornazheva, B. P.	Need satisfaction, motivation, and well-being in the work organizations of a former eastern bloc country: A cross-cultural study of self-determination [32]	*Personality and Social Psychology Bulletin*	841
2002	Luthans, F.	The need for and meaning of positive organizational behavior [7]	*J. of Organizational Behavior*	994
2003	Sonnentag, S.	Recovery, work engagement, and proactive behavior: A new look at the interface between nonwork and work [33]	*Journal of Applied Psychology*	855
2004	Baard, P. P., Deci, E. L., and Ryan, R. M.	Intrinsic need satisfaction: A motivational basis of performance and well-being in two work settings [34]	*Journal of Applied Social Psychology*	821
2005	Salanova, M., Agut, S., and Peiro, J. M.	Linking organizational resources and work engagement to employee performance and customer loyalty: The mediation of service climate [35]	*Journal of Applied Psychology*	996
2006	Schaufeli, W. B., Bakker, A. B., and Salanova, M.	The measurement of work engagement with a short questionnaire—A cross-national study [36]	*Educational and Psychological Measurement*	2386
2007	Luthans, F., Avolio, B. J., Avey, J. B., and Norman, S. M.	Positive psychological capital: Measurement and relationship with performance and satisfaction [37]	*Personnel Psychology*	1220
2008	Bakker, A. B., Schaufeli, W. B., Leiter, M. P., and Taris, T. W.	Work engagement: An emerging concept in occupational health psychology [38]	*Work and Stress*	824
2009	Schaufeli, W. B., Bakker, A. B., Van Rhenen, W.	How changes in job demands and resources predict burnout, work engagement, and sickness absenteeism [39]	*Journal of Organizational Behavior*	817
2010	Bakker, A. B., and Bal, P. M.	Weekly work engagement and performance: A study among starting teachers [40]	*J. of Occupational and Organizational Psychology*	433
2011	Christian, M. S., Garza, A. S., and Slaughter, J. E.	Work engagement: A quantitative review and test of its relations with task and contextual performance [41]	*Personnel Psychology*	958
2012	Bakker, A. B., Tims, M., and Derks, D.	Proactive personality and job performance: The role of job crafting and work engagement [42]	*Human Relations*	331
2013	Karatepe, O. M.	High-performance work practices and hotel employee performance: The mediation of work engagement [43]	*International J. of Hospitality Management*	162
2014	Abbas, M., Raja, U., Darr, W., and Bouckenooghe, D.	Combined effects of perceived politics and psychological capital on job satisfaction, turnover intentions, and performance [44]	*Journal of Management*	181
2015	Barnes, C. M., Lucianetti, L., Bhave, D. P., and Christian, M. S.	You wouldn’t like me when I’m sleepy: leaders’ sleep, daily abusive supervision, and work unit engagement [45]	*Academy of Management Journal*	134
2016	Baron, R. A., Franklin, R. J., and Hmieleski, K. M.	Why entrepreneurs often experience low, not high, levels of stress: The joint effects of selection and psychological capital [46]	*Journal of Management*	129
2017	Di Fabio, A.	Positive healthy organizations: Promoting well-being, meaningfulness, and sustainability in organizations [47]	*Frontiers in Psychology*	111
2018	Tawfik, D. S., Profit, J., Morgenthaler, T. I., Satele, D. V., Sinsky, C. A., Dyrbye, L. N., Tutty, M. A., West, C. P., and Shanafelt, T. D.	Physician burnout, well-being, and work unit safety grades in relationship to reported medical errors [48]	*Mayo Clinic Proceedings*	114
2019	Schaufeli, V. B., Shimazu, A., Hakanen, J., Salanova, M., and De Witte, H.	An ultra-short measure for work engagement the UWES-3 validation across five countries [49]	*European Journal of Psychological Assessment*	68
2020	Qing, M., Asif, M., Hussain, A., and Jameel, A.	Exploring the impact of ethical leadership on job satisfaction and organizational commitment in public sector organizations: The mediating role of psychological empowerment [50]	*Review of Managerial Science*	24

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
