# Peer review of "Positive Organizational Psychology: A Bibliometric Review and Science Mapping Analysis"

_ijerph, 2021, doi:10.3390/ijerph18105222_

Round 1

Reviewer 1 Report

This is a very interesting and worthy project and generated a great deal of data, the latter of which is valuable and also challenging in terms of its organization.

This is an ambitious purpose: "Therefore, this work aims to: (1) analyze temporal research trends and the most in- 78
fluential articles according to the citations received; (2) identify the scientific categories 79
involved; (3) explore the distribution of countries, institutions, and authors, as well as 80
their collaborative relationships; (4) discover which are the most influential journals; and 81
(5) understand what the core research topics have been in the field of POP, after conduct- 82
ing a review of the articles published in the database Web of Science Core Collection 83
(WoSCC), through the use of bibliometric methodology and science mapping. "

It seems like this could be an entire book that has the space for laying out the context, comparing it to other studies and the history of comparable areas of research using this method, the history of the method, the implications of the findings, etc. As it is, it seems to veer in many intriguing directions. So I would like to see a bit more meaning-making, structure, and direction to guide me as a reader as to what to pay attention to. I understand the need for objectivity but framing is always necessary and can be helpful in providing focus.

For the most part the article is well-written, although there are a few places that require attention. Here is an example in lines 352-360:

Concerning the analysis of citations carried out, as an objective measure of perfomance, value, recognition, influence, and impact of an investigation and its researchers [51], it should be noted that their amount has been increasing in the first two periods of publication. However, the exponential increase in the third period was not matched by an increase in the number of citations received, but rather the contrary. Between 2012 and 356
2020, this number has been declining dramatically, going from 34.26 citations per work in 2012 to 1.19 in 2020. 

There is evidence for and against the greater likelihood of the oldest works to be cited [52]... 

I don't understand the immediately preceding sentence, (I did click on the citation (52) but the article cited does not seem to be a primary source--can you just tell us briefly about the evidence, whether here or in a more appropriate place?) and the paragraph before that is also somewhat unclear and yet seems to have major implications. Is the field on a downswing? What might we know of patterns from other studies using this method? Maybe the authors don't want to speculate here, but in any event, maybe the paragraph should come later. It was very interesting, but also distracting considering the placement.

The present tense in the very first paragraph referring to events in the past was unexpected and I am not sure if it was deliberate style choice.

Valuable information! My students could use it to help them guide lit reviews.

Author Response

I have attached the reply to the review report (Reviewer 1)

Reviewer 2 Report

Comments "Positive Organizational Psychology: A bibliometric review and science mapping analysis":

  • It is not logical to use so many search equations. This is done to obtain the largest number of articles and to be able to compare them with other bases. When you include (*) in a term, you succinctly assume that the search is not defined. Nor is it clear why you choose the work of Donaldson and Ko (19) as the key to determine the search, besides being from 2010. Why not others who have updated the topic?
    - In Figure 1 the numbers do not match
    - What is the contribution of this work to the field of research beyond offering some data on POP drivers? What relevance does it have for a journal that focuses on the publication of scientific and technical information on the impacts of natural phenomena and anthropogenic factors on the quality of our environment, the interrelationships between environmental health and the quality of life, as well as the socio- cultural, political, economic, and legal considerations related to environmental stewardship, environmental medicine, and public health?
    - In this work there is no progress in the research on POP, but only reflects a static photo of the research carried out. It does not really include any significant contribution that allows the researcher to start from the conclusions of this to continue with the development of his study. It would be convenient, perhaps in another job, to make a systematic review of the articles that are directly linked to POP.

Author Response

I have attached the reply to the review report (Reviewer 2)

Reviewer 3 Report

This is a really nice article for scholars hoping to get a better understanding of the history and scope of positive organizational psychology as a field. The methods were clearly articulated and the findings were edifying. 

Additional thought could be given to the implications of the findings presented. There are many potential future research directions that emerge from these findings, so it would be helpful to have the authors indicate how this contribution enhances the field. 

Methodologically this is a strong paper that gives a nice high level overview of POP. I would recommend a revision so that they can dedicate more space to discussing the implications of their findings and emerging gaps in the POP literature. While the methods and results are clearly presented and edifying, there is little in the introduction or conclusion that helps the reader understand the contribution the paper makes to the field beyond summary. 

Author Response

I have attached the reply to the review report (Reviewer 3)

Round 2

Reviewer 2 Report

The work continues to need a development of the previous literature that articulates the discussions.

Author Response

Dear Reviewer 2

Following your suggestion:

  1. In this regard, for example, in the introduction (lines 67 -73) a paragraph was included with the results of the searches of all the review articles on the topic of Positive Psychology and Positive Organizational Psychology.

The need to develop an article with a bibliometric methodology was also justified in view of the lack of studies of this type in this area (lines 74-84).

  1. On the other hand, we have also included suggestions in the discussion based on our study results:

(Lines 418-419) It is important that academics and experts who research in the field of POP make their work visible to the scientific community.

(Lines 439-440) It would be interesting to the scientific community to increase international collaboration between countries and research groups to consolidate POP.

  1. Finally, we have related updated literature results about Work and Organizational Psychology with our results in POP in the discussion section:

(Lines 362-364) The same trend has been observed in the study by Sott et al. [12] in their review of 100 years of scientific evolution in the field of Work and Organizational Psychology

(Lines 443-444) This is in line with the findings of Soot et al [12] in defining them as new themes unrelated to earlier periods in the development of Work and Organizational Psychology.

Therefore, we consider that these modifications allow us to adequately articulate the development of the discussion.

Yours sincerely

Ángel Solanes.
